# Surgical Outcome of Refixation versus Exchange of Dislocated Intraocular Lens: A Retrospective Cohort Study

**DOI:** 10.3390/jcm9123868

**Published:** 2020-11-28

**Authors:** Young In Shin, Un Chul Park

**Affiliations:** 1Department of Ophthalmology, Seoul National University College of Medicine, Seoul 03080, Korea; poohsyi@hanmail.net; 2Retinal Degeneration Research Laboratory, Seoul National University Hospital Biomedical Research Institute, Seoul 03080, Korea

**Keywords:** intraocular lens dislocation, refixation, intraocular lens exchange, scleral fixation

## Abstract

We compared the surgical outcomes and complications of refixation vs. exchange of dislocated intraocular lenses (IOLs) in patients who underwent transscleral suture fixation combined with pars plana vitrectomy for the treatment of IOL dislocation. A total of 83 eyes (*n* = 83 patients) with postoperative follow-up of ≥6 months were evaluated: 40 received refixation of dislocated IOL (refixation group) while 43 received IOL exchange (exchange group) treatment. Treatment outcomes, including best-corrected visual acuity (BCVA), spherical equivalent, corneal cylinder, intraocular pressure (IOP), central macular thickness (CMT), and corneal endothelial cell density (ECD), and postoperative complications were retrospectively reviewed. BCVA improvement at 6 months after surgery was comparable between the groups. Postoperative decrease in corneal ECD was significantly greater in the exchange group than in the refixation group, but no significant differences were found in spherical equivalent, corneal cylinder, IOP, or CMT changes. The exchange group experienced significantly more frequent postoperative vitreoretinal complications, such as retinal detachment, choroidal effusion, cystoid macular edema, and secondary epiretinal membrane, than the refixation group. Without any reason to extract the dislocated IOL, reuse of the dislocated IOL would be a better surgical option for transscleral suture fixation to protect corneal endothelial cells and prevent postoperative vitreoretinal complications.

## 1. Introduction

Dislocation of the intraocular lens (IOL) into the vitreous cavity is a serious late complication after cataract extraction and IOL implantation and is reported to occur in up to 2.0% of patients [1,2,3]. In-the-bag IOL dislocation usually occurs several years after uneventful cataract surgery and is thought to result from progressive loosening of the zonulae [4]. Risk factors for zonular dehiscence include pseudoexfoliation syndrome [5,6], trauma [7,8], connective tissue disorders [9], uveitis [7,8], retinitis pigmentosa [10], axial myopia [8,10], and a history of vitreoretinal surgery [7,8,11]. In contrast, out-of-the-bag IOL dislocation usually occurs in the early postoperative period after complicated cataract surgery [11].

For both types of IOL dislocation, IOL placement in the lens capsule or in the sulcus is almost impossible due to severe defects in or absence of capsular or zonular structure, and the IOL should be fixed to the sclera or iris. In general, there are two surgical options for the management of a dislocated IOL: one is rescue and refixation of the dislocated IOL to the sclera and the other is exchanging the dislocated IOL for a new IOL with fixation to the sclera or iris. Methods to fixate the posterior chamber IOL include transscleral suture fixation [12,13], sutureless intrascleral fixation [14,15], and retropupillary fixation of the iris-claw IOL [16,17]. The incidence of IOL dislocation in the vitreous cavity will grow continually in the future as life expectancy increases, and the determination of the best surgical approach in each patient will be of greater importance in clinical practice. However, it is still unknown whether reuse and exchange of dislocated IOLs have different postoperative outcomes and complications, although it is assumed that reuse is less traumatic. In this study, we compared the surgical outcomes and complications of refixation vs. exchange of dislocated IOLs in patients who underwent transscleral suture fixation combined with pars plana vitrectomy for the treatment of IOL dislocation.

## 2. Materials and Methods

### 2.1. Participants

In this retrospective study, we reviewed the medical records of consecutive patients who underwent transscleral suture fixation combined with pars plana vitrectomy for the treatment of IOL dislocation at the Seoul National University Hospital from September 2016 to August 2019. The exclusion criteria were as follows: (1) presence of a concurrent vitreoretinal complication (e.g., retinal detachment, macular diseases); (2) combined surgery with penetrating keratoplasty or glaucoma surgery; (3) history of underlying corneal disease (e.g., corneal laceration, bullous keratopathy, or Fuchs dystrophy); (4) presence of any ocular comorbidity that can affect visual acuity (e.g., glaucoma); and (5) a follow-up period <6 months. This study was approved by the Institutional Review Board of the Seoul National University Hospital (IRB no. 2010-148-1167), and all investigations adhered to the tenets of the Declaration of Helsinki. Informed consent was obtained from all patients before surgery.

### 2.2. Surgical Procedures

All surgeries were performed by a single surgeon (UCP). All patients underwent a standard 23-gauge three-port vitrectomy. After the induction of posterior vitreous detachment, core vitrectomy was performed to free the dislocated IOL from the vitreous and the posterior capsular bag complex, especially for in-the-bag IOL dislocation. In some cases, 1–2 mL of perfluorocarbon liquid was temporally injected into the vitreous cavity to protect the macula and to facilitate the handling of the dislocated IOL. Peripheral vitrectomy was performed to remove as much vitreous as possible, especially at the inferior and superior vitreous base where transscleral suture fixation would be performed. The peripheral retina was carefully examined under scleral indentation for any retinal break. The dislocated IOL was evaluated for reusability. In cases with optic opacification, defect in haptics, instability of optic–haptic junction, or inappropriate design of haptics for suture fixation (e.g., plate haptic), the dislocated IOL was removed and a new IOL was implanted (exchange group). If the dislocated IOL was intact, it was reused for transscleral suture fixation (refixation group). In the exchange group, dislocated IOLs were grasped with intraocular forceps and lifted up to the anterior chamber, which was maintained with an ophthalmic viscosurgical device (1.5% sodium hyaluronate) throughout the surgical procedure. The IOL was cut in half using an IOL cutter and extracted through the nasal or temporal clear corneal incision.

After pars plana vitrectomy, transscleral suture fixation of the IOL was performed as previously described by Kim et al. [18]. A conjunctival incision was made superiorly and inferiorly. A long-curved double-armed 10-0 polypropylene needle was passed through the sclera approximately 2.0 mm posterior to the limbus at 12 o’clock, and a 26-gauge hollow needle was passed from the opposite site at 6 o’clock. After docking the solid needle into the tip of the hollow needle, the two needles were removed together. After that, two 2.75 mm clear corneal incisions were made at 4 and 10 o’clock. The polypropylene suture thread was pulled through the 10 o’clock corneal incision with a Sinskey hook and cut in two. Then, one end of the suture thread was withdrawn through the 4 o’clock incision using McPherson forceps, while the other was left through the 10 o’clock incision. In the refixation group, each haptic of the dislocated IOL was externalized through the two corneal incisions, and the cut suture threads were tensely tied to each haptic. In the exchange group, a new foldable 3-piece hydrophobic acrylic IOL (AR40e; AMO, Santa Ana, CA, USA) was inserted and tied in the same manner. In both groups, after the sutured haptics were placed behind the iris, the sutures were tightened by pulling the polypropylene thread taut. Centration of the IOL was carefully checked, and the sutures were tied to close the sclerotomy sites at 6 and 12 o’clock. The polypropylene suture knots were left long to prevent conjunctival erosion and suture exposure and were buried under the conjunctiva. Conjunctivas were repaired with 7-0 polyglactin (Vicryl, Ethicon, Somerville, NJ, USA), and the 23-gauge sclerotomy sites were also repaired if leakage was noted. Stromal hydration was administered at both edges of the clear corneal incisions to facilitate healing.

### 2.3. Clinical Outcome Measures

Patient demographic data such as age, sex, laterality of the operated eye, and time elapsed since cataract surgery were collected. All patients underwent a comprehensive ophthalmic examination at baseline and during the postoperative period: best-corrected visual acuity (BCVA) measurement using a Snellen chart, intraocular pressure (IOP), slit lamp examination, fundus examination, refractive error, corneal endothelial cell density (ECD), spectral domain optical coherence tomography (OCT), and ultrawide field retinal imaging (Optos 200TX; Optos PLC, Scotland, UK). Refractive status was examined using autokeratometry (KR-8900, Topcon, Tokyo, Japan), and spherical and cylindrical powers were converted to the spherical equivalent. ECD was measured using a noncontact confocal microscope (Noncon Robo, Konan, Japan). Central macular thickness was obtained from the central 1 mm subfield in the macular thickness map of the spectral domain OCT (Cirrus HD-OCT; Carl Zeiss Meditec, Jena, Germany). At postoperative visits, ocular complications were monitored and recorded, including retinal detachment, choroidal effusion, endophthalmitis, cystoid macular edema (CME), IOP elevation, suture exposure, corneal decompensation, secondary epiretinal membrane, hypotonic maculopathy, optic capture, and redislocation of the IOL.

### 2.4. Statistical Analyses

BCVA was converted to the logarithm of the minimum angle of resolution (logMAR) values for statistical analysis. Statistical analysis was performed using SPSS 22.0 software for Windows (SPSS Inc., Chicago, IL, USA). Comparisons between the two groups were made using the Mann–Whitney *U* test and chi-square test. Pre- and postoperative parameters were compared using the Wilcoxon signed-rank test. The data are presented as means ± standard deviations, and *p* < 0.05 was considered statistically significant. A post hoc power analysis was conducted to determine the power for the sample size of this study, and 85% power ≥80% was achieved for the main outcomes (BCVA and ECD at 6 months after surgery).

## 3. Results

Eighty-three eyes from 83 patients were included in this study: 40 and 43 eyes were categorized into the refixation and exchange groups, respectively. The baseline characteristics and preoperative clinical data are summarized in Table 1. The mean time duration from the IOL implantation to dislocation in the refixation group was significantly longer than in the exchange group (15.4 ± 11.1 vs. 9.6 ± 6.8 years, *p* = 0.005), but there were no significant differences in other baseline characteristics between the groups. In the exchange group, indications for removal of dislocated IOLs were opacification of the optic in 19 eyes (41.3%), inappropriate design of haptics in 13 eyes (28.3%), defects in haptics in 10 eyes (21.7%), and optic–haptic junction instability in one eye (2.2%).

Postoperative outcomes at 6 months after surgery are summarized in Table 2. BCVA change at 6 months after surgery was −0.31 ± 0.58 in the refixation group and −0.14 ± 0.77 in the exchange group, and there was no significant intergroup difference (*p* = 0.347). BCVA at 6 months after surgery was better in the refixation group than in the exchange group (0.24 ± 0.25 vs. 0.59 ± 0.78; *p* = 0.015). Corneal ECD decreased from 2032.8 ± 596.1 cells/mm^2^ at baseline to 1895.7 ± 691.3 cells/mm^2^ at 6 months in the refixation group (*p* = 0.100) and from 2080.7 ± 588.1 cells/mm^2^ to 1481.8 ± 553.1 cells/mm^2^ in the exchange group (*p* < 0.001). Corneal ECD at 6 months after surgery in the exchange group was significantly lower than that in the refixation group (*p* = 0.041), and the decrease in corneal ECD was greater in the exchange group (−599.0 ± 413.3 vs. −137.1 ± 344.9 cells/mm^2^, *p* < 0.001). Other postoperative parameters, including spherical equivalent, amount of astigmatism, and central macular thickness (CMT), did not differ between the groups. Postoperative changes in BCVA and CMT during follow-up are shown in Figure 1. The mean BCVA significantly differed at 1, 3, and 6 months after surgery, while the mean CMT did not differ during follow-up.

There were no intraoperative complications in either group, while postoperative complications occurred in 16 of 40 eyes (40.0%) in the refixation group and 26 of 43 eyes (60.5%) in the exchange group (*p* = 0.018). These complications are listed in Table 3. IOP elevation occurred more frequently in the refixation group than in the exchange group, but the difference was marginally significant (*p* = 0.063). All cases with IOP elevation were successfully managed with medical treatment. Among the eyes with IOL capture, one eye in the exchange group underwent a repositioning procedure, but the others showed resolution of capture spontaneously or after pupillary dilation in the supine position. Corneal decompensation occurred in three eyes in the exchange group and one eye in the refixation group.

Development of vitreoretinal complications was more frequent in the exchange group than in the refixation group (*n* = 12/43 (27.9%) vs. *n* = 2/40 (5.0%); *p* = 0.005). Choroidal effusion in four eyes in the exchange group spontaneously resolved during follow-up. CME developed only in five eyes in the exchange group, and complete resolution was seen after topical use of bromfenac sodium 0.1% (Bronuck; Senju Pharmaceutical Co., Ltd., Osaka, Japan) in three eyes, intravitreal injection of triamcinolone acetonide in one eye, and sub-Tenon injection of dexamethasone in one eye. Rhegmatogenous retinal detachment occurred in one eye that underwent the exchange of a polymethyl methacrylate IOL with a large incision and was successfully treated with repeated vitrectomy and gas tamponade. Re-dislocation of the IOL occurred in one eye due to suture breakage in the exchange group at 22 months after surgery and was treated with refixation of the IOL without further complications. Hypotonic maculopathy occurred in only two eyes in the refixation group, of which one eye was treated with intravitreal gas injection, while the other showed spontaneous resolution. Excluding the potentially vision-threatening complications, such as corneal decompensation and vitreoretinal complications, BCVA improvement in the exchange group was significant (0.57 ± 0.62 to 0.29 ± 0.47; *p* = 0.003).

## 4. Discussion

In this retrospective study, surgical outcomes and postoperative complications were compared between the refixation and exchange of dislocated IOLs during transscleral suture fixation combined with pars plana vitrectomy to treat intravitreally dislocated IOL. Although the visual outcomes were comparable, refixation of the dislocated IOLs was more protective of corneal endothelial cells and resulted in less frequent vision-threatening vitreoretinal complications during the postoperative period, suggesting that refixation is a better surgical option whenever a dislocated IOL is undamaged.

The surgical outcomes of the present study were generally consistent with findings from previous studies that compared refixation and exchange of dislocated IOLs. Between the groups, there was no significant difference in postoperative BCVA and BCVA change during the 6 months, as reported previously [19,20,21,22,23]. Both groups experienced visual improvement, but it was significant only in the refixation group in the present study. Surgically induced astigmatism was not evaluated in this study, but postoperative astigmatism was comparable between the groups. Recent retrospective studies reported a significant decrease in corneal ECD after both refixation and exchange of a dislocated IOL, and reported no significant differences in corneal ECD at 6 months after surgery or in the ECD decrease between the groups [19,20,24]. However, ECD change in the present study was remarkably consistent with a prospective randomized trial by Kristianslund et al. that compared IOL reposition by scleral suture and exchange with retropupillary fixation of an iris-claw IOL [22]. In their study, the decrease in corneal ECD was only significant in the exchange group, which showed both a statistically significantly greater amount of ECD loss and a lower ECD at 6 months when compared with the reposition group.

There are several considerations when determining the surgical technique for the management of IOL dislocation, but the first is to check the conformational status and material of the optic. Dislocated IOLs with opacification of the optic, structural defects in the haptics or optic–haptic junctions, or inappropriate design of the haptics for fixation should be exchanged for a new IOL, while those with intact structure and appropriate design for scleral fixation may be considered for reuse. In particular, dislocated IOLs with a rigid optic material such as polymethyl methacrylate require a large incision up to 6 mm for removal, potentially resulting in greater postoperative astigmatism or more frequent complications. Exchange of an IOL carries a risk of vitreous prolapse, corneal endothelium damage, iris trauma, and retinal damage, especially during the extraction of the dislocated IOL [25]. A higher incidence of postoperative complications and greater loss of ECD in the exchange group in this study confirms this possibility, suggesting that the reuse of a dislocated IOL would be a better surgical option if it is undamaged and appropriate for fixation. Preoperative corneal status is also important, and eyes with decreased ECD should be considered for reuse of a dislocated IOL when available to prevent postoperative corneal decompensation.

In this study, the exchange group experienced significantly more frequent vitreoretinal complications during the postoperative follow-up. Differences in incidence of vitreoretinal complications between the two techniques have not been reported in the literature. In particular, transient choroidal effusion, CME, and secondary epiretinal membrane were observed exclusively in the exchange group. This may be attributed to subclinical inflammation in the posterior segment resulting from prolonged surgical time and increased surgical trauma on the uveal tissue during the exchange of IOLs [26,27], as overt intraocular inflammation after surgery was not more persistent in the exchange group.

On the other hand, reuse of dislocated IOLs for transcleral suture fixation also has some potential disadvantages. The haptics of IOLs dislocated in-the-bag may be distorted by the contraction of the capsular bag for a long time [7]. Minute structural alterations of haptics may result in tilting of the IOL after suture fixation, particularly when the changes are asymmetric between arms. In addition, while grasping and pulling out the haptics of dislocated IOLs to make a tie on them, further mechanical stress may be applied on the haptics or optic–haptic junctions. In particular, if dislocated IOLs are reused with a sutureless intrascleral fixation technique, haptics and junctions would be under greater mechanical stretch than before [28]. In a small case series, Baba et al. preferred the exchange of dislocated IOLs to their reuse based on their finding that three of six patients treated with refixation showed separation of the optic–haptic junction or IOL tilting during follow-up, while nine patients treated with exchange did not [23]. However, the sample size of the study was very small compared with other studies. Judicious case selection for reuse of a dislocated IOL and careful manipulation during surgery would prevent conformational changes in the IOL during follow-up, which was not observed in this study. In the present study, IOP elevation during follow-up was relatively more frequent in the refixation group, although it was well controlled medically in all cases. This does not seem to be associated with the optic capture and resultant pupillary block considering its comparable incidence between the groups, and the reason for the more frequent IOP elevation in the refixation group remains to be explained.

The main limitations of this study were its retrospective design and relatively short postoperative follow-up period. In addition, treatment outcomes of the refixation group in this study were not compared with sutureless intrascleral fixation or IOL exchange by retropupillary fixation of an iris-claw IOL, which are widely used currently. However, in this study, based on the same technique of transscleral suture fixation, a pure comparison between the reuse and exchange of dislocated IOLs was available for surgical outcomes and postoperative complications.

## 5. Conclusions

In conclusion, without any reason to extract a dislocated IOL, reuse of the IOL would be a preferred option to exchanging the IOL for scleral fixation in order to protect corneal endothelial cells and minimize the risk of postoperative vitreoretinal complications. Judicious case selection and careful manipulation of dislocated IOLs are required. Further studies with long-term follow-up would be helpful to determine the best surgical approach for dislocated IOLs in terms of the reusability and technique for fixation of IOLs.

## Figures and Tables

**Figure 1 jcm-09-03868-f001:**
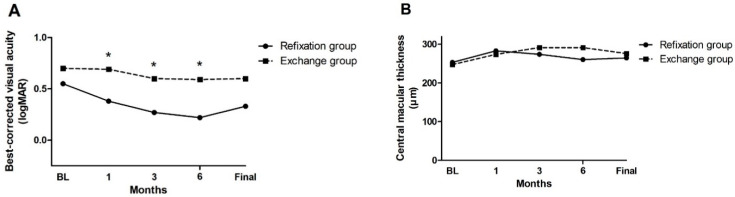
Changes in the mean best-corrected visual acuity (**A**) and central macular thickness (**B**) during follow-up in the refixation and exchange groups. Asterisks indicate significant difference.

**Table 1 jcm-09-03868-t001:** Baseline characteristics of the participants in the Refixation and Exchange group.

	Refixation Group (*n* = 40)	Exchange Group (*n* = 43)	*p*-Value
Age (years)	65.3 ± 14.0	61.9 ± 13.7	0.271
BCVA (logMAR)	0.55 ± 0.63	0.73 ± 0.69	0.243
IOP (mmHg)	17.7 ± 6.6	16.7 ± 5.1	0.418
Spherical equivalent (SE)	4.4 ± 7.4	7.3 ± 5.9	0.091
Astigmatism (cylinder diopter)	−1.6 ± 1.7	−1.4 ± 1.5	0.503
Time duration from IOL implantation (years)	15.4 ± 11.1	9.6 ± 6.8	0.005
ECD (cells/mm^2^)	2032.8 ± 596.1	2080.7± 588.1	0.794
CMT (μm)	253.2 ± 87.0	247.3 ± 84.3	0.788
Axial length (mm)	25.4 ± 3.1	25.0 ± 2.3	0.547
Postoperative follow-up (months)	16.2 ± 10.9	16.6 ± 12.7	0.891

BCVA, best-corrected visual acuity; logMAR, logarithm of the minimum angle of resolution; IOP, intraocular pressure; IOL, intraocular lens; ECD, endothelial cell density.

**Table 2 jcm-09-03868-t002:** Treatment Outcomes at 6 months after Surgery.

	Refixation Group (*n* = 40)	Exchange Group (*n* = 43)	*p*-Value
BCVA (logMAR)	0.24 ± 0.25	0.59 ± 0.78	0.015
BCVA change* (logMAR)	−0.31 ± 0.58	−0.14 ± 0.77	0.347
IOP (mmHg)	16.2 ± 4.4	17.2 ± 4.1	0.371
Spherical equivalent (SE)	−2.0 ± 1.7	−1.1 ± 2.3	0.245
Astigmatism (cylinder diopter)	−1.8 ± 1.6	−1.8 ± 1.5	0.981
ECD (cells/mm^2^)	1895.7 ± 691.3	1481.8 ± 553.1	0.041
ECD change * (cells/mm^2^)	−137.1 ± 344.9	−599.0 ± 413.3	<0.001
CMT (μm)	260.3 ± 61.9	291.3 ± 137.2	0.309
CMT change * (μm)	−16.2 ± 87.5	64.9 ± 161.2	0.072

BCVA, best-corrected visual acuity; logMAR, logarithm of the minimum angle of resolution; IOP, intraocular pressure; ECD, endothelial cell density; CMT, central macular thickness. * Changes within the group (before and 6 months after surgery).

**Table 3 jcm-09-03868-t003:** Postoperative complications of the Refixation and Exchange group.

Complications	Refixation Group (*n* = 40)	Exchange Group (*n* = 43)	*p*-Value
IOP increase	7 (17.5%)	2 (4.7%)	0.063
IOL capture	3 (7.5%)	7 (16.3%)	0.234
Pupillary block attack	0	1 (2.3%)	0.344
Suture exposure	1 (2.5%)	2 (4.7%)	0.620
Corneal decompensation	1 (2.5%)	3 (7.0%)	0.341
Prolonged intraocular inflammation	3 (7.5%)	8 (18.6%)	0.151
Vitreoretinal complications			
Choroidal detachment	0	4 (9.3%)	0.044
Retinal detachment	0	1 (2.3%)	0.344
Central macular edema	0	5 (11.6%)	0.024
IOL re-dislocation into vitreous cavity	0	1 (2.3%)	0.344
Endophthalmitis	0	0	N/A
Secondary epiretinal membrane	0	1 (2.3%)	0.344
Hypotonic maculopathy	2 (5.0%)	0	0.136
Any vitreoretinal complications	2 (5.0%)	12 (27.9%)	0.005

IOL, intraocular lens; IOP, intraocular pressure.

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
