# Peer review of "Surgical Outcome of Refixation versus Exchange of Dislocated Intraocular Lens: A Retrospective Cohort Study"

_jcm, 2020, doi:10.3390/jcm9123868_

Round 1
Reviewer 1 Report
Amazing work by the authors.
Author Response
Thank you very much for your supportive comment.
Reviewer 2 Report
Dear colleagues,
congratulations on a really nice manuscript, which incorporates many individually different patients, showing very interesting results, nicely presented. I would recommend minor changes just to improve English language, and correcting few grammar errors. I think the article will be interesting for the Journal reader.
Best Regards
Author Response
Thank you very much for your supportive comment. This manuscript underwent English editing by native editor.
Reviewer 3 Report
The authors present a highly relevant study and provide sound data to a frequently recurrent clinical dilemma: explant an IOL or resuture.
The article is well written and a large number of cases is presented. This will provide much needed clinical guidance.
I do recommend a post-hoc power calculation be added to see whether enough patients were looked at to answer the question but the p-value suggests this already. Calculator at https://clincalc.com/stats/Power.aspx
Please phrase the research question as a hypothesis in the last paragraph of the introduction as is proper practice in hypothesis driven research (anything else is difficult to get funding for).
Author Response
The authors present a highly relevant study and provide sound data to a frequently recurrent clinical dilemma: explant an IOL or resuture.
The article is well written and a large number of cases is presented. This will provide much needed clinical guidance.
I do recommend a post-hoc power calculation be added to see whether enough patients were looked at to answer the question but the p-value suggests this already. Calculator at https://clincalc.com/stats/Power.aspx
→ Thank you very much for valuable comment. We conducted a post-hoc power analysis to determine the power for the sample size as you recommended and power over 80 % was achieved for the main outcome measurements such as ECD and BCVA at 6 months after surgery.
Please phrase the research question as a hypothesis in the last paragraph of the introduction as is proper practice in hypothesis driven research (anything else is difficult to get funding for).
→ I appreciate your constructive advice, and we phrased the main hypothesis in the last paragraph of the introduction as “However, it is still unknown whether reuse and exchange of dislocated IOL have different postoperative outcome and complications, although it was assumed that reuse is less traumatic.”
Reviewer 4 Report
The authors have compared the outcomes and complications of refixation vs. exchange of dislocated IOLs.
The paper contains all the necessary information. Everything is well-described. The data received in this study support the author’s conclusion. The Discussion presents high quality as well.
I have no concerns about this manuscript. It may help the surgeons in their decisions in the cases of IOL dislocation.
Author Response

(The authors gave the same response as above.)
